# Multi-Robot Connected Fermat Spiral Coverage

**Primary Keywords:** *Multi-Agent Planning; Robotics*

## Abstract

We introduce Multi-Robot Connected Fermat Spiral (MCFS), a novel algorithmic framework for Multi-Robot Coverage Path Planning (MCPP) that adapts Coverage Fermat Spiral (CFS) from the computer graphics community to multi-robot coordination for the first time. MCFS uniquely enables the orchestration of multiple robots to generate coverage paths that contour around arbitrarily shaped obstacles, a feature notably lacking in traditional methods. Our framework not only enhances area coverage and optimizes task performance, particularly in terms of makespan, for workspaces rich in irregular obstacles but also addresses the challenges of path continuity and curvature critical for non-holonomic robots by generating smooth paths without decomposing the workspace. MCFS solves MCPP by constructing a graph of isolines and transforming MCPP into a combinatorial optimization problem, aiming to minimize the makespan while covering all vertices. Our contributions include developing a unified CFS version for scalable and adaptable MCPP, extending it to MCPP with novel optimization techniques for cost reduction and path continuity and smoothness, and demonstrating through extensive experiments that MCFS outperforms existing MCPP methods in makespan, path curvature, coverage ratio, and overlapping ratio. Our research marks a significant step in MCPP, showcasing the fusion of computer graphics and automated planning principles to advance the capabilities of multi-robot systems in complex environments.

## 1 Introduction

In the evolving landscape of multi-robot systems, the efficiency and effectiveness of Multi-Robot Coverage Path Planning (MCPP) (Almadhoun et al. 2019) remain pivotal in a myriad of applications, ranging from environmental monitoring (Collins et al. 2021) to search-and-rescue operations (Song et al. 2022) in complex workspaces. Traditional methodologies, such as cellular decomposition (Latombe and Latombe 1991; Acar et al. 2002) and grid-based methods (Gabriely and Rimon 2001; Hazon and Kaminka 2005), have laid a solid foundation for understanding and navigating the challenges inherent in these tasks. However, as the complexity of environments and the demand for more efficient coverage increase, there is a growing need for innovative strategies that can adeptly handle workspaces rich in irregular obstacles with both high precision and adaptability.

This paper introduces a novel algorithmic framework, called Multi-Robot Connected Fermat Spiral (MCFS), which revolutionizes MCPP by building upon the principles of Coverage Fermat Spiral (CFS) (Zhao et al. 2016) from the computer graphics community. This represents the first application of leveraging CFS to solve MCPP challenges in automated planning and robotics, showcasing a unique interdisciplinary fusion. MCFS stands out for its unique ability to coordinate multiple robots in generating contour-like coverage paths, elegantly adapting to the intricacies of arbitrary-shaped obstacles—a characteristic not typically addressed by traditional methods. This contouring ability of MCFS results in not only more organic and less segmented coverage but also enhanced task efficiency in both time and energy expenditure by balancing the path costs across multiple robots, as indicated by the makespan (Zheng et al. 2010) metric.

Besides task efficiency, a key challenge in MCPP is managing the deceleration and sharp turns required by non-holonomic robots. Traditional methods (Lu et al. 2023; Vandermeulen, Groß, and Kolling 2019), often focused on minimizing path turns, are constrained to rectilinear workspaces and rely on decomposing the area into rectangles. This approach is less effective in arbitrary-shaped environments. In contrast, the essence of our MCFS framework lies in its global coverage strategy, conceptualizing the paths as a series of interconnected spirals that seamlessly integrate the movements of multiple robots. This strategy results in smooth covering paths without the need for decomposition, inherently accounting for path curvature—a vital factor for efficient robotic navigation.

Drawing inspiration from the original application of CFS (Zhao et al. 2016) in additive manufacturing (Gibson et al. 2021), our MCFS framework innovatively adapts CFS to tackle the MCPP problem, which generates continuous and smooth coverage paths by converting a set of equidistant contour-parallel isolines into connected Fermat spirals. MCFS first constructs a graph of isolines (Sec. 3), associating each vertex with an isoline and connecting it to associated vertices of adjacent isolines. It then reduces the MCPP problem to Min-Max Rooted Tree Cover (MMRTC), a combinatorial optimization problem that finds a forest of trees to cover all vertices of the graph while minimizing the makespan (Sec. 4). Our framework is versatile, allowing coverage paths to start from arbitrary starting points as required in MCPP, and optimizes the distribution of the cov-

erage of both multiple whole isolines and segments of an isoline among multiple robots, showcasing an innovative approach to effectively managing the makespan, curvature, and path continuity for each robot.

**Key Contributions:** (1) We propose a unified version of CFS that standardizes the stitching of adjacent isolines, allowing for customized priorities in selecting stitching points and providing scalability and ease of adaptation to MCPP by enabling coverage paths to start from any given initial robot positions. (2) We demonstrate how our MCFS framework extends this unified version of CFS to MCPP and effectively solves the corresponding MMRTC problem. (3) We introduce two optimization techniques: one that adds edges between non-adjacent but connectable pairs of isolines to expand the solution space and another that refines the MMRTC solution for balanced path costs and reduced overlap in multi-robot coverage. (4) We present extensive experimental results validating the superiority of our MCFS framework over state-of-the-art MCPP methods in metrics of makespan, path curvature, coverage ratio, and overlapping ratio, showcasing its effectiveness in diverse coverage scenarios.

## 2  Related Work

We categorize existing Single-Robot Coverage Path Planning (CPP) and MCPP methods into grid-based, cellular decomposition, and global methods. We refer interesting readers to (Tomaszewski 2020) for a more detailed taxonomy.

**Grid-Based Methods:** Grid-based methods abstract workspaces into square grids (Hazon and Kaminka 2005; Kapoutsis, Chatzichristofis, and Kosmatopoulos 2017; Tang, Sun, and Zhang 2021), allowing for the application of various graph algorithms. One prominent method, Spanning Tree Coverage (STC) (Gabriely and Rimon 2001), constructs a minimum spanning tree and then generates circumnavigating paths on the tree to cover the workspace. STC-based MCPP methods (Hazon and Kaminka 2005; Tang and Ma 2023) work by finding a set of trees that jointly visit all vertices and assigning each robot a path that circumnavigates a tree. While convenient, the complexity of optimally solving grid-based MCPP grows exponentially in the workspace size and the number of robots.

**Cellular Decomposition Methods:** These methods decompose the workspace into sub-regions by detecting geometrical critical points, such as trapezoid (Latombe and Latombe 1991) and Morse (Acar et al. 2002) decomposition. CPP methods generate zigzag paths in these sub-regions for coverage (Choset 2000; Wong and MacDonald 2003), and MCPP methods connect and assign these sub-regions, filled with zigzag paths, to robots for cooperative coverage (Rekleitis et al. 2008; Mannadiar and Rekleitis 2010; Karapetyan et al. 2017). Additionally, some research optimizes the direction of the zigzag paths for single robots (Oksanen and Visala 2009; Bochkarev and Smith 2016). Although efficient, these methods are less suitable for obstacle-rich or non-rectilinear workspaces due to their reliance on geometric partitioning.

**Global Methods:** Global CPP methods directly generate paths to cover the workspace without decomposing it. They fall into two types: the first type generates separate paths that contour around obstacles (Yang et al. 2002), and the second type generates a closed path, including Spiral Path (Ren, Sun, and Guo 2009) and CFS (Zhao et al. 2016) that are notable for their continuous and smooth paths. CFS paths are especially convenient as their entry and exit points are adjacent, facilitating the integration of multiple paths. A recent paper has built a CFS path based on an exact geodesic distance field to cover a terrain surface (Wu et al. 2019). However, to our knowledge, there are no global methods yet developed for MCPP.

## 3  Connected Fermat Spiral (CFS)

In this section, we present our unified version of CFS, an adaptation of the original CFS (Zhao et al. 2016) concept. The original CFS employs a two-phase process to transform a set of equidistant isolines into a closed path that covers an input polygon workspace. It utilizes a graph structure, where vertices represent individual isolines and edges connect vertices whose respective isolines have adjacent segments. Initially, the original CFS identifies a set of "pockets"—connected components on the spanning tree of the graph. The first phase transforms the isolines within each pocket into a *Fermat spiral* (Wiki 2023a), and the second phase stitches these isolated Fermat spirals to construct the final, connected Fermat spiral by traversing the pockets using the graph edges.

Our unified version of CFS modifies the graph construction of isolines and consolidates the original two-phase process into a singular, cohesive operation for the CPP problem. The primary modification in our approach lies in the stitching phase. Rather than explicitly identifying pockets and then stitching isolated Fermat spirals, we employ a unified procedure for every pair of stitchable isolines. Instead of explicitly identifying pockets and then stitching the resultant isolated Fermat spirals, our method integrates a unified process that simultaneously addresses both the conversion of isolines within a pocket into Fermat spirals and the interconnection of these spirals. This integrated process is applied to every pair of stitchable isolines, effectively merging the conversion and stitching phases. By traversing a rooted spanning tree of the graph, the same connected Fermat spiral is obtained as the original CFS.

The advantage of our unified CFS approach is twofold. Firstly, it enhances scalability, facilitating the incorporation of diverse utilities within the framework. Secondly, it simplifies the extension of CFS to MCPP, as will be elaborated in Sec. 4. By integrating and streamlining the conversion and stitching processes, our approach improves the efficiency and versatility of CFS for complex CPP challenges.

### 3.1  Constructing Isolines and the Isograph

We describe our approach for generating layered isolines from a given polygon workspace to be covered and building the isograph. The polygon is enclosed by its boundary, consisting a set of interior boundary polylines that represent obstacles and an exterior boundary polyline.

**Generating Layered Isolines:** The procedure starts by sampling a 2D mesh grid of points within the polygon. A distance field is built for these points, representing their shortest

distance to the polygon boundary (encompassing both the interior obstacle boundary polylines and the exterior boundary polyline). We denote the distance step size between isolines as $l$, and the largest distance to the polygon boundary among all points as $l_{max}$. We then use the *Marching Squares* algorithm (Wiki 2023b) to generate layered isolines for each layer $i = 1, 2, ..., \lfloor l_{max}/l \rfloor$. This ensures that the distance between each point in the layer-$i$ isoline and the polygon boundary is $l \times i$. The last step resamples equidistant points along each isoline, maintaining a consistent distance of $l$ between adjacent points.

**Building the Isograph:** We define *isograph* of the layered isolines as an undirected graph $G = (V, E)$, where $V$ is the set of *isovertices*, each associated with a unique isoline. For ease of reference, we let $I_v$ and $L_v$ denote the isoline associated with any $v \in V$ and its respective layer. Similar to the original CFS (Zhao et al. 2016), we define a *connecting segment set* $O_{u,v}$ for any pair of isovertices $u, v \in V$ in adjacent layers (i.e., $|L_u - L_v| = 1$) as:

$$O_{u \to v} = \{\mathbf{p} \in I_u \mid \forall z \in V, d(\mathbf{p}, I_v) < d(\mathbf{p}, I_z) \land L_z = L_v\} \quad (1)$$

where $d(\mathbf{p}, I)$ denotes the distance between point $\mathbf{p}$ and isoline $I$. Unlike the original CFS which directly constructs an undirected edge $(u, v)$ if $O_{u \to v}$ is nonempty, we also consider $O_{v \to u}$ for edge construction. This consideration provides flexibility in traversing the isograph in any order and from any root isovertex in the CFS context. It also avoids adding edges $(u, v)$ where the respective isolines $I_u$ and $I_v$ are separated by multiple isolines, as such pairs may be unsuitable for stitching in the CPP context (see Sec. 3.4). Therefore, we define a set $O_{u,v}$ of *stitching tuples* for any $u, v \in V$ in adjacent layers as:

$$O_{u,v} = \{(\mathbf{p}, \mathbf{q}) \in O_{u \to v} \times O_{v \to u} \mid \mathbf{p} = \mathcal{C}_u(\mathbf{q}) \land \mathbf{q} = \mathcal{C}_v(\mathbf{p})\} \quad (2)$$

where $\mathcal{C}_u(\mathbf{p})$ denotes the nearest point along isoline $I_u$ to point $\mathbf{p}$. Subsequently, an undirected edge $(u, v)$ is formed for any $u, v \in V$ in adjacent layers with a nonempty $O_{u,v}$. Each tuple $(\mathbf{p}, \mathbf{q}) \in O_{u,v}$ serves as a candidate stitching tuple to connect isolines $I_u$ and $I_v$ by stitching $\mathbf{p}$ to $\mathbf{q}$ and $\mathcal{B}_u(\mathbf{p})$ to $\mathcal{B}_v(\mathbf{q})$, where $\mathcal{B}_u(\mathbf{p})$ denotes the point preceding $\mathbf{p}$ along isoline $I_u$ in counterclockwise order.

While the original CFS assigns a weight of $|O_{u \to v}|$ to each edge to retain a low-curvature path when determining the isograph traversal order for connecting isolated Fermat spirals, we currently leave the edge weight definition application-specific and will explicitly address this objective in Sec. 3.3 for every stitching operation.

## 3.2 Unifying the CFS algorithm

We detail our unified version of CFS in Alg. 1, which takes as input an isograph $G$ and an entry point $\mathbf{p}_0$. The algorithm starts by identifying the isovertex $r$ containing $\mathbf{p}_0$ (line 1) as the root for obtaining the Depth-First-Search (DFS) traversal edges of $G$. Line 2 initializes the CFS path $\pi$ to be constructed and the set $U$ to record points already used for stitching the isolines. The main loop then iterates over the DFS edges (line 3) and stitches the corresponding pair of isolines for each edge (line 5-6). Specifically, a stitching tuple $(\mathbf{p}, \mathbf{q})$ is selected via any selector (line 5). For any isovertex $v \in V$, we use $I_v(\mathbf{p})$ to denote the counterclockwise path along isoline $I_v$ starting at $\mathbf{p}$ and ending at $\mathcal{B}_v(\mathbf{p})$. This path

---

**Algorithm 1:** Unified Version of CFS

**Input:** isograph $G$, entry point $\mathbf{p}_0$
1   $r \leftarrow$ the isovertex of $G$ containing $\mathbf{p}_0$
2   $\pi \leftarrow I_r(\mathbf{p}_0)$,   $U \leftarrow \emptyset$
3   **for** $(u, v) \in$ DFS traversal edges of $G$ from $r$ **do**
4     remove any $(\mathbf{p}, \mathbf{q})$ from $O_{u,v}$ where $\mathbf{p} \in U$ or $\mathbf{q} \in U$
5     $(\mathbf{p}, \mathbf{q}) \leftarrow f(O_{u,v})$     $\triangleright$ by any selector $f$ in Sec. 3.3
6     stitch $I_v(\mathbf{p})$ into $\pi$ by stitching $\mathbf{p}$ to $\mathbf{q}$ and $\mathcal{B}(\mathbf{p})$ to $\mathcal{B}(\mathbf{q})$
7     $U \leftarrow U \cup \{\mathbf{p}, \mathbf{q}\}$
8   **return** $\pi$

---

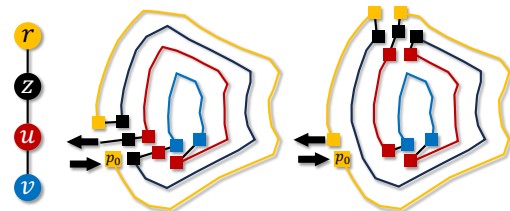

Figure 1: The unified version of CFS. Colored squares represent the stitching tuples. From left to right: The input isograph, the path resulting from the CFS selector, and the path resulting from the MCS selector.

segment is then stitched into $\pi$ using the selected stitching tuple (line 6). The set $U$ is updated to include these newly selected stitching tuples (line 7). Following the interation over all DFS edges, the final path $\pi$ is constructed to stitch together all isolines and completely cover the given polygon.

## 3.3 Stitching Tuple Selector

We now propose three stitching tuple selectors, each designed to select an appropriate stitching tuple $\mathbf{s}$ from a given set $O_{u,v}$ for connecting isolines $I_u$ and $I_v$. Fig. 1 demonstrates an example of these selectors.

**Random Selector:** The random selector $f_{\text{rnd}}$ randomly selects a stitching tuple from the set $O_{u,v}$.

**Connected Fermat Spiral (CFS) Selector:** The CFS selector $f_{\text{cfs}}$ aligns our unified version of CFS with the original CFS. It attempts to select a stitching tuple from $O_{u,v}$ for $(u, v) \in E$ that is adjacent to the previously selected stitching tuple of $(z, u) \in E$ or $(z, v) \in E$. Either $(z, u)$ or $(z, v)$, with its stitching tuple already selected by $f_{\text{cfs}}$, will be visited before $(u, v)$ in the DFS traversal (line 3). Assuming that $(z, u)$ is visited first with the selected stitching tuple $(\mathbf{p}', \mathbf{q}') \in O_{z,u}$, $f_{\text{cfs}}$ then checks for $\mathbf{s} = (\mathbf{p}, \mathbf{q})$ in $O_{u,v}$ where $\mathcal{B}(\mathbf{p}) = \mathbf{q}'$. If such a tuple exists, it is selected for $(u, v)$; otherwise, the first tuple in $O_{u,v}$ is selected.

**Minimum Curvature Stitching (MCS) Selector:** The MCS selector $f_{\text{mcs}}$ iterates through $O_{u,v}$ to identify the stitching tuple $\mathbf{s} = (\mathbf{p}, \mathbf{q})$ that minimizes the curvature difference $\Delta \kappa(\mathbf{s})$ before and after stitching, defined as:

$$\Delta \kappa(\mathbf{s}) = \sum_{\mathbf{p} \in \mathbf{s}} [\kappa_\pi(\mathbf{p}) - \kappa_{I_u}(\mathbf{p})] \quad (3)$$

where $\kappa_\pi(\mathbf{p})$ and $\kappa_{I_u}(\mathbf{p})$ denote the curvatures at any point $\mathbf{p}$ on the new stitched path $\pi$ using $\mathbf{s}$ and on the original isoline $I_u$, respectively. Formally, the MCS selector is defined as $f_{\text{mcs}}(O_{u,v}) = \arg \max_{\mathbf{s} \in O_{u,v}} \Delta \kappa(\mathbf{s})$.

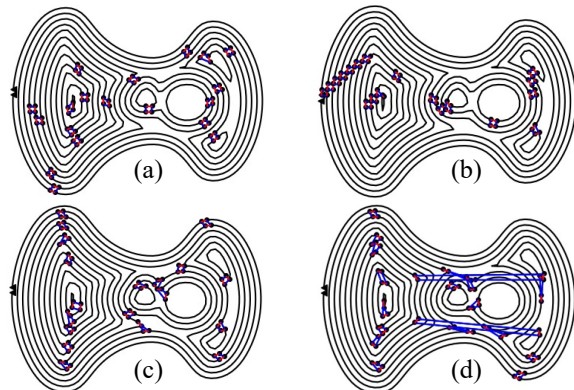

Figure 2: CFS paths resulting from the (a) random selector, (b) CFS selector, (c) MCS selector, and (d) MCS selector with $O_{u\rightarrow v}$. Black triangles, blue lines, and red lines are the entry and exit points, the stitching path segments, and the removed isoline segments after stitching.

## 3.4 Case Study: Unified vs Original CFS

We discuss the necessity of modification in the construction of the isograph edge set of our unified version of CFS in the CPP context. Unlike the original CFS (Zhao et al. 2016) that uses a set $O_{u\rightarrow v}$ in (Eqn. (1)) for edge set construction and always starts traversal from the lowest-layer isovertices, our unified CFS defines a more versatile set $O_{u,v}$ (Eqn. (2)). This modification addresses the requirement in CPP (and MCPP) for starting a coverage path from an arbitrary given point $\mathbf{p}_0$, as accommodated by Alg. 1. Our unified CFS starts the graph traversal from isovertex $r$, whose respective isoline contains $\mathbf{p}_0$, without the restriction of $r$ being the lowest-layer isovertex. Consequently, valid stitching tuples may not exist for edge construction if only the single-directional tuples from layer $i$ to layer $i+1$ are considered as in the original CFS. Moreover, an isovertex $u$ with a local innermost isoline may find a nonempty $O_{u\rightarrow v}$ for any isovertex $v$ with $L_v = L_{u+1}$, recognizing $(u, v)$ as an edge, which potentially introduces path overlapping. Fig. 2-(d) exemplifies two such cases where two local innermost layer-6 isolines are stitched to a layer-7 isoline separated by other isolines, a scenario effectively managed in our unified CFS but problematic in using the original CFS definitions. The other figures in Fig. 2 further visualize the three stitching tuple selectors in Sec. 3.3. Fig. 2-(b) visualizes the staircase-like stitching scheme in the original CFS (Zhao et al. 2016) using the CFS selector. Fig. 2-(c) shows that the MCS selector always picks the stitching tuples at high-curvature positions in order to minimize the curvature.

# 4 Multi-Robot CFS Coverage

In this section, we present our MCFS framework for solving MCPP. MCFS computes multiple trees from an input isograph, each corresponding to a different robot, and then applies CFS on each tree to compute individual coverage paths. In Sec. 4.1, we detail the CFS-based formulation of MCPP and introduce its reduction to Min-Max Rooted Tree Cover (MMRTC) (Even et al. 2004; Tang and Ma 2023). Since there can still be unnecessary repetition in the coverage paths resulting from an optimal MMRTC solution, we present two optimization techniques, isograph augmentation in Sec. 4.2 and solution refinement in Sec. 4.3, aiming to further enhance the MCPP solution.

## 4.1 Problem Formulation

We present our problem formulation of MCPP that facilitates the extension of CFS. The problem of MCPP is to find a set $\Pi = \{\pi_i\}_{i\in I}$ of coverage paths for a set $I$ of robots that minimizes the makespan (i.e., the maximum path cost). Following existing literature (Zheng et al. 2010; Tang, Sun, and Zhang 2021), we assume that each robot starts and ends at a given position, corresponding to a pair of adjacent entry and exit points in the CFS context. Formally, the objective of MCPP is minimizing the makespan $\tau$, represented as:

$$\min_{\Pi} \tau = \min_{\Pi=\{\pi_i\}_{i\in I}} \max\{c(\pi_1), c(\pi_2), ..., c(\pi_{|I|})\}. \quad (4)$$

When using CFS to generate each coverage path in $\Pi$, the path length is linear in $|\pi|$ and thus the cost of any path $\pi$ can be evaluated as $c(\pi) = |\pi|$, given that each isoline in CFS contains equidistant points (as detailed in Sec.3.1). For an isograph $G = (V, E)$, each $v \in V$ is assigned a weight $w_v = |I_v|$, representing the number or points in isoline $I_v$. Consequently, the cost of any tree $T \subseteq G$ is $c(T) = \sum_{v\in V(T)} w_v$. The MMRTC problem parallels MCPP in its aim to find a makespan-minimizing set of rooted trees, where each graph vertex is covered by at least one tree. Given a graph $G = (V, E)$ and a set $R = \{r_i\}_{i\in I} \subseteq V$ of root isovertices for the robots, the objective of MMRTC is defined as:

$$\min_{\mathcal{T}=\{T_i\}_{i\in I}} \max\{c(T_1), c(T_2), ..., c(T_{|I|})\} \quad (5)$$

where each $T_i \in \mathcal{T}$ is a tree rooted at $r_i$. Let $V(T)$ and $E(T)$ denote the vertex set and edge set of any tree $T$, respectively. The solution set $\mathcal{T}$ must satisfy $v \in \bigcup_{i\in I} V(T_i)$ to ensure coverage of every $v \in V$. Since CFS stitches each isoline $I_v$ of $v \in V(T_i)$ to construct the coverage path $\pi_i \in \Pi$, we have $c(\pi_i) = |\pi_i| = \sum_{v\in V(T_i)} |I_v| = c(T_i)$. Hence, for any isograph $G$ and set $R$ of root isovertices for the robots, the heuristic values in Eqn. (4) and Eqn. (5) are identical under CFS, effectively reducing MCPP to MMRTC.

We employ the Mixed Integer Programming (MIP) model proposed in (Tang and Ma 2023) to solve MMRTC optimally. The optimal set of trees obtained is then used to produce coverage paths by applying our unified CFS (Alg. 1) on each tree. Fig. 4-(a) and (b) illustrate a 2-tree MMRTC instance and its crresponding solution.

## 4.2 Optimization: Isograph Augmentation

In Sec. 3.1, the isograph building process considers each edge only for two isolines in adjacent layers. This process, while efficient, often results in a sparse graph structure in the isograph and thus an undesirable MMRTC solution where certain isovertices are repetitively covered by multiple trees. One common example of such repetition appears for a *cut isovertex*, defined as a vertex whose removal increases the number of connected components in the graph. Such repetitions become more common as the number of trees (robots)

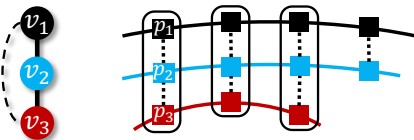

Figure 3: Left: The augmented isograph with original edges (solid lines) and an augmented edge (dashed line). Right: Three sequences of stitching tuples (black boxes) for $O_{v_1,v_3}$.

increases or when tree roots are clustered, thereby leading to increased makespan and reducing the overall quality of MCPP solutions. To mitigate this issue, we propose to augment the sparse isograph with additional edges connecting isovertices in non-adjacent layers. This augmentation aims to reduce the sparsity of the isograph and allow MMRTC trees to explore new routes for joint coverage, thereby reducing repetitions and balancing tree costs.

The augmentation of an isograph $G = (V, E)$ operates by adding a set $E^{\#}$ of augmented edges, defined as:
$$E^{\#} = \{(u,v) \mid \forall u, v \in V, 2 \le d_G(u,v) \le \delta\} \qquad (6)$$
where $d_G(\cdot, \cdot)$ denotes the graph distance between any two isovertices in $G$, and $\delta$ is a hyperparameter that sets the augmentation level. The set $E^{\#}$ is then used to update $G$ by setting $E = E \cup E^{\#}$. For edges in $E^{\#}$, stitching tuples are constructed differently from those in adjacent layers described in Sec. 3.1. Without loss of generality, we consider an edge $(v_1, v_{k+1}) \in E^{\#}$ and its shortest path $(v_1, v_2, ..., v_{k+1})$ in the original $G$ (i.e., each segment $(v_i, v_{i+1})$ is part of E and $k$ is the graph distance between $v_1$ and $v_k$). The set $O_{v_1,v_{k+1}}$ comprises all pairs of $\mathbf{p}_1$ on the isoline of $v_1$ and $\mathbf{p}_{k+1}$ on the isoline of $v_{k+1}$ that can be feasibly connected, forming valid stitching tuples $(\mathbf{p}_1, \mathbf{p}_{k+1})$. Such points are connectable iff they form a sequence of consecutive stitching tuples $(\mathbf{p}_1, \mathbf{p}_2) \in O_{v_1,v_2}, \ldots, (\mathbf{p}_k, \mathbf{p}_{k+1}) \in O_{v_k,v_{k+1}}$, which ensures that the straight-line segment between the pair does not intersect more than $k - 1$ or any obstacles within the workspace. Given that the distance between adjacent isolines is set as $l$ (Sec. 3.1), we assign a weight $w_e = l \times k$ to each augmented edge $e = (u, v) \in E$ with a layer difference of $k$ (i.e., $|L_u - L_v| = k$), which approximates the additional path cost incurred by any tree containing the augmented edge. The cost of any tree $T$ is thus updated to $c(T) = \sum_{v \in V(T)} w_v + \sum_{e \in E(T)} w_e$ in MMRTC solving.

### 4.3 Optimization: MMRTC Solution Refinement

Despite isograph augmentation reducing isovertex repetitions in the optimal MMRTC solution, two bottlenecks persist in achieving a better MCPP solution. The first bottleneck results from certain isovertex repetitions that remain unresolved by augmentation alone, notably when multiple robots share the same root isovertex or multiple trees use the same vertex. To tackle this, we implement the PAIRWISEISOVERTICESSPLITTING (PIS) function, designed to disperse the coverage of the isoline of an isovertex with repetitions amount multiple robots. The second bottleneck arises from the limitation of an optimal MMRTC solution in balancing tree costs when isoline traversing costs vary significantly. To tackle this, we propose the ADDIMPROVINGREPETITION (AIR) function that selectively adds an isovertex from a higher-cost tree to a lower-cost tree and uses PIS to split this reassigned isovertex, effectively redistributing isoline coverage between trees. We call the additional repetition introduced by AIR an *improving repetition*. Both PIS and AIR are crucial in refining the MMRTC solution: PIS directly addresses the issue of shared isovertices, while AIR strategically adjusts coverage load distribution to balance costs among the trees, enhancing the overall MCPP solution.

**Pseudocode:** Alg. 2 (Lines 1-14) details the process for refining an MMRTC solution using the AIR and PIS functions. The process starts with initializing the MMRTC solution $\mathcal{T}^*$ to be returned and the set $U$ of isovertices already used for PIS [Line 1]. It then builds the set $M$ of isovertices with repetitions from the input solution $\mathcal{T}$ [Line 2]. If $M$ is empty, the process calls AIR to potentially add an improving repetition to $M$ [Line 3]. The process then arranges $M$ into a max-heap to prioritize splitting the isovertex with the largest number of occurrences among different trees [Line 4]. It then iterates over $M$ and to address each isovertex with repetitions one at a time [Lines 5-14]. Each $u$ popped from $M$ [Line 6] identifies the set $\mathcal{T}_s$ of all trees containing $u$ [Line 7]. PIS then evaluates the splitting of $u$ with each adjacent, unused neighbor $v$ [Line 10], returning

---

**Algorithm 2:** MMRTC Solution Refinement

**Input:** isograph $G = (V, E)$, optimal MMRTC solution $\mathcal{T}$
1  $\mathcal{T}^* \leftarrow \mathcal{T}, \; U \leftarrow \emptyset$
2  $M \leftarrow \{u \in V \mid \sum_{T \in \mathcal{T}} |\{u\} \cap V(T)| > 1\}$
3  ADDIMPROVINGREPETITION$(\mathcal{T}, M, U)$ if $M = \emptyset$
4  max-heapify $M$ ordered by the number of occurrences
5  **while** $M \neq \emptyset$ **do**
6     $u \leftarrow M.pop()$
7     $\mathcal{T}_s \leftarrow \{T \in \mathcal{T} \mid u \in V(T)\}$
8     $h^* \leftarrow +\infty, \mathcal{T}_s^* \leftarrow \mathcal{T}_s, \mathcal{T}_n \leftarrow \mathcal{T}/\mathcal{T}_s$
9     **for** $(u,v) \in \{(u,v) \in E \mid v \notin U\}$ **do**
10      $h, \mathcal{T}_s \leftarrow$ PAIRWISEISOVERTICESSPLITTING$(\mathcal{T}_s, u, v)$
11      set $h^*$ to $h$ and $\mathcal{T}_s^*$ to $\mathcal{T}_s$ if $h < h^*$
12    $\mathcal{T} \leftarrow \mathcal{T}_n \cup \mathcal{T}_s^*, U \leftarrow U \cup \{u, v\}, M \leftarrow M/\{v\}$
13    set $\mathcal{T}^*$ to $\mathcal{T}$ if its evaluated makespan is smaller
14    ADDIMPROVINGREPETITION$(\mathcal{T}, M, U)$ if $M = \emptyset$
15 **return** $\mathcal{T}^*$
16 **Function** ADDIMPROVINGREPETITION$(\mathcal{T}, M, U)$**:**
17    sort $\mathcal{T}$ by the evaluated costs in ascending order
18    $P \leftarrow \{(u,v) \mid u, v \in V(\mathcal{T}[-1])/U \wedge deg(u, \mathcal{T}[-1]) = 1\}$
19    **for** $((u,v), T) \in P \times \mathcal{T}$ **do**
20      **if** $u \in \{b \in V(T) \mid \exists (b, x) \in E, x \notin V(T)\}$ **then**
21        $T \leftarrow (V(T) \cup \{u\}, E(T) \cup \{(u,v)\}), M \leftarrow M \cup \{u\}$
22        **return**

23 **Function** PAIRWISEISOVERTICESSPLITTING$(\mathcal{T}_s, u, v)$**:**
24    $k \leftarrow |\mathcal{T}_s|, h^* \leftarrow +\infty, \mathcal{T}_s^* \leftarrow \mathcal{T}_s$
25    **for** $\mathbf{c} = (\mathbf{s}_1, \mathbf{s}_2, ..., \mathbf{s}_k) \in O_{u,v}^k$ **do**
26      $Z \leftarrow$ a set of $k$ new isovertices split from $u, v$ by stitching $I_u, I_v$
       using $\mathbf{s}_1, \mathbf{s}_2, ..., \mathbf{s}_k$    ▷ see Fig. 4-(d)
27      **for** $(T, z) \in zip(\mathcal{T}_s, Z)$ **do**
28        $E_T \leftarrow \{(u, x) \mid x \in \mathcal{N}_T(u)\} \cup \{(v, x) \mid x \in \mathcal{N}_T(v)\}$
29        $E_T' \leftarrow \{(z, x) \mid (\cdot, x) \in E_T\}$    ▷ see Fig. 4-(c)
30      $\mathcal{T}_s' \leftarrow \{(V(T) \cup \{z\}/\{u, v\}, E(T) \cup E_T'/E_T) \mid T \in \mathcal{T}_s\}$
31      $h \leftarrow \sigma(\{c(T)\}_{T \in \mathcal{T}_s'})$
32      mark $\mathcal{T}_s'$ as nonadjacent if $\exists T \in \mathcal{T}_s$ and $(z, x) \in E_T', |O_{z,x}| = 0$
33      **for** $(z, x) \in E_T'$ with an empty $O_{z,x}$ **do**
34        add the distance between $I_z, I_x$ to $h$
35      set $h^*$ to $h$ and $\mathcal{T}_s^*$ to $\mathcal{T}_s'$ if $h < h^*$
36    **return** $h^*, \mathcal{T}_s^*$

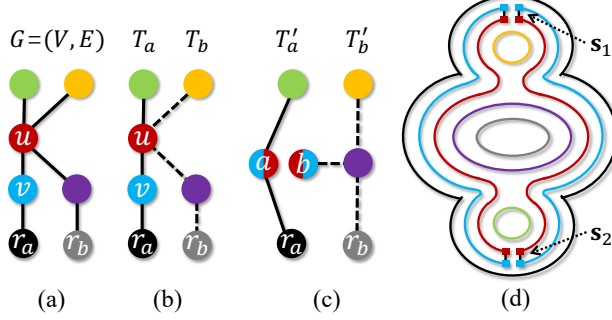

$G=(V,E)$  $T_a$  $T_b$  $T'_a$  $T'_b$

(a)  (b)  (c)  (d)

Figure 4: Pairwise isovertices splitting from $u, v$ into $a, b$ at stitching tuples $\mathbf{s}_1, \mathbf{s}_2$. (a) Isograph $G$. (b)(c) Two trees of $G$ (in dashed and solid lines, respectively) before and after the splitting. (d) The layered isolines, each corresponding to the isovertex highlighted in the same color.

its heuristic value $h$ and the post-split tree set $\mathcal{T}_s$ to update $\mathcal{T}_s^*$ if $h < h^*$ [Line 11]. The best tree set $\mathcal{T}_s^*$ for splitting $u$ with the smallest heuristic value is then integrated into the MMRTC solution $\mathcal{T}$ [Line 12], with a subsequent update to $\mathcal{T}^*$ [Line 13]. If $M$ is empty, the process calls AIR again to potentially add an additional improving repetition [Line 14]. As every iteration records isovertices used for PIS in $U$ and AIR only adds unused isovertices, Alg. 2 terminates after at most $|V|/2$ iterations since two new isovertices are added to $U$ in Line 12 in each iteration.

**Add Improving Repetition (AIR):** The AIR function (Lines 16-22) identifies an isovertex from the highest-cost tree in the tree set $\mathcal{T}$ and adds it to a low-cost tree. It first sorts $\mathcal{T}$ by tree costs in ascending order [Line 17] and then builds a set $P$ of isovertex pairs from the highest-cost tree $\mathcal{T}[-1]$ [Line 18]. Note that $P$ contains only those pairs where both isovertices are not in $U$ (hence, unused for PIS), and the first isovertex $u$ in each pair must be a leaf (having a degree of 1), making it an ideal candidate to be split from $\mathcal{T}[-1]$. By iterating through each tree in $\mathcal{T}$, ordered by cost [Line 19], the first isovertex $u$ is validated for addition to the first tree [Line 21] where $u$ is a neighbor of any isovertex in that tree [Line 20]. Once an isovertex is added to a low-cost tree, the function terminates [Line 22], ensuring that only one improving repetition is added to $M$ per AIR call.

**Pairwise Isovertices Splitting (PIS):** The PIS function (Lines 23-36) splits isovertex $u$ with repetitions and its neighbor $v$ into a set of new isovertices, each integrated into a corresponding tree in $\mathcal{T}_s$. The function iterates through all possible mappings $\mathbf{c} = (\mathbf{s}_1, \mathbf{s}_2, ..., \mathbf{s}_k)$ from the splitting tuples in $O_{u,v}$ to the $k$ trees in $\mathcal{T}_s$ (through the $k$-th Cartesian power set of $O_{u,v}$) [Line 25] and finds the best mapping with the smallest heuristic value $h^*$. For each $\mathbf{c}$, it splits $u$ and $v$ into a set $Z$ of $k$ new isovertices, each representing a segment of isolines $I_u$ and $I_v$ connected via the corresponding splitting tuples [Line 26]. The function then constructs edge subsets $E_T$ and $E'_T$ for each tree $T \in \mathcal{T}_s$ [Lines 27-29], where $E_T$ comprises edges connected to $u$ or $v$ to be removed from $T$ [Line 28] and $E'_T$ comprises new edge to be added to $T$ [Line 29]. Notably, the stitching tuple set $O_{z,x}$

of each new edge $(z, x)$ is conveniently obtained by admitting all valid stitching tuples for isoline $I_z$ and the new combined isoline $I_x$. The function builds the new tree set for $\mathbf{c}$ [Line 30] and calculates the standard deviation of the tree costs as its heuristic value [Line 31], aiming for cost balance. The function then validates for $\mathbf{c}$ that each such $O_{z,x}$ set is non-empty, otherwise marking the new tree set as non-adjacent [Line 32] and penalizing the heuristic value $h$ for any empty $O_{z,x}$ by adding the distance between $I_z$ and $I_x$ [Line 34]. Finally, the heuristic value $h$ is compared with $h^*$ for potential updates to $\mathcal{T}_s^*$ [Line 35]. Note that if the final $\mathcal{T}_s^*$ [Line 15] is marked nonadjacent in Line 32 and used in CFS, then shortest paths are inserted between isoline pairs with empty $O$ sets to compensate for missing valid stitching tuples. Fig. 4-(b) shows isovertex $u$, with repetitions of two trees, split into two new isovertices via PIS in Fig. 4-(c).

## 5 Empirical Evaluation

This section presents our experimental results on a 3.49 GHz Apple® M2 CPU laptop with 16GB RAM. Our code will be publicly available upon acceptance of this paper.

**Setup:** The MMRTC MIP model in Sec. 4.1 is solved using the Gurobi solver (Gurobi Optimization, LLC 2023) with a

| Selectors | char-I | char-C | char-A | char-P | char-S | 2-torus | office |
|---|---|---|---|---|---|---|---|
| random | 2.824 | 0.924 | 1.228 | 2.095 | 1.084 | 1.070 | 12.93 |
| CFS | 1.306 | 0.747 | 0.848 | 1.724 | 0.887 | 0.819 | 11.77 |
| MCS | 1.269 | 0.782 | 0.874 | 1.277 | 0.960 | 0.969 | 8.289 |

Table 1: Curvature comparison between stitching tuple selectors in the unified version of CFS for single-robot CPP.

| | Method | char-I | char-C | char-A | char-P | char-S | 2-torus | office |
|---|---|---|---|---|---|---|---|---|
| | robots | 2 | 2 | 3 | 4 | 5 | 6 | 9 |
| Makespan (↓) | TMC | 99.94 | 136.3 | 87.75 | 75.19 | 62.51 | 133.7 | 154.0 |
| | TMSTC* | 91.33 | 117.9 | 84.35 | 50.63 | 56.41 | 113.9 | 238.1 |
| MCFS NONE | 132.3 | 179.8 | 75.4 | 106.8 | 50.46 | 174.2 | 291.0 |
| MCFS +REF | **69.74** | 125.7 | 63.44 | 52.86 | 50.46 | 108.9 | 213.4 |
| MCFS +AUG | 85.37 | 106.3 | 63.14 | 48.23 | 46.26 | 87.86 | 155.5 |
| MCFS +BOTH | 70.75 | **105.0** | **63.14** | **35.13** | **36.04** | **80.73** | **141.2** |
| Curvature | TMC | 2.541 | 3.433 | 7.482 | 6.115 | 5.011 | 3.341 | 8.459 |
| | TMSTC* | 2.476 | 1.801 | 2.655 | 2.869 | 2.259 | 1.335 | 2.117 |
| MCFS NONE | 1.129 | 0.776 | **0.950** | 0.970 | 1.050 | 1.299 | 1.192 |
| MCFS +REF | 2.512 | 0.842 | 0.981 | 1.184 | 1.050 | 1.357 | 1.737 |
| MCFS +AUG | **0.972** | **0.758** | 1.047 | **0.828** | **0.787** | 1.070 | **1.087** |
| MCFS +BOTH | 1.026 | 0.795 | 1.047 | 1.428 | 1.068 | **1.064** | 1.352 |
| Coverage | TMC | 86.8% | 87.6% | 88.4% | 88.0% | 85.8% | 91.5% | 89.2% |
| | TMSTC* | 90.6% | 92.4% | **91.0%** | **90.2%** | 91.2% | 93.7% | **91.3%** |
| MCFS NONE | 91.1% | 92.4% | 89.5% | 89.4% | 91.9% | **94.6%** | 91.2% |
| MCFS +REF | 91.1% | 92.4% | 89.4% | 89.4% | 91.9% | 94.5% | 91.1% |
| MCFS +AUG | **91.1%** | **92.5%** | 89.4% | 89.4% | **91.9%** | 94.5% | 91.1% |
| MCFS +BOTH | 91.0% | 92.4% | 89.4% | 89.4% | 91.8% | 94.5% | 91.1% |
| Overlapping | TMC | 8.76% | 7.76% | **5.59%** | 7.89% | 18.8% | 15.8% | 15.3% |
| | TMSTC* | 8.12% | 6.25% | 9.37% | 13.1% | 16.5% | 15.5% | 17.1% |
| MCFS NONE | 82.6% | 5.46% | 5.91% | **62.5%** | 6.79% | 86.6% | 50.2% |
| MCFS +REF | **6.50%** | **5.44%** | 5.92% | 7.95% | **6.79%** | 25.0% | 24.0% |
| MCFS +AUG | 22.4% | 6.41% | 6.75% | 22.2% | 7.48% | 20.0% | 24.5% |
| MCFS +BOTH | 7.27% | 6.25% | 6.63% | 10.8% | 7.41% | **9.62%** | **13.1%** |
| Runtime | TMC | 0.25s | 1.26s | 0.97s | 0.33s | 76.0s | 30.4m | 31.2m |
| | TMSTC* | 1.21s | 1.78s | 1.77s | 1.02s | 2.70s | 8.22s | **27.9s** |
| MCFS NONE | **0.24s** | **0.38s** | **0.44s** | **0.29s** | **0.31s** | 1.57s | 30.1m |
| MCFS +REF | 8.59s | 11.7s | 8.60s | 5.08s | 0.60s | 39.8s | 33.1m |
| MCFS +AUG | 0.34s | 0.60s | 0.85s | 0.46s | 0.60s | 13.9m | 30.2m |
| MCFS +BOTH | 7.13s | 12.5s | 20.0s | 7.89s | 15.6s | 15.2m | 37.5m |

Table 2: Solution quality for different MCPP algorithms.

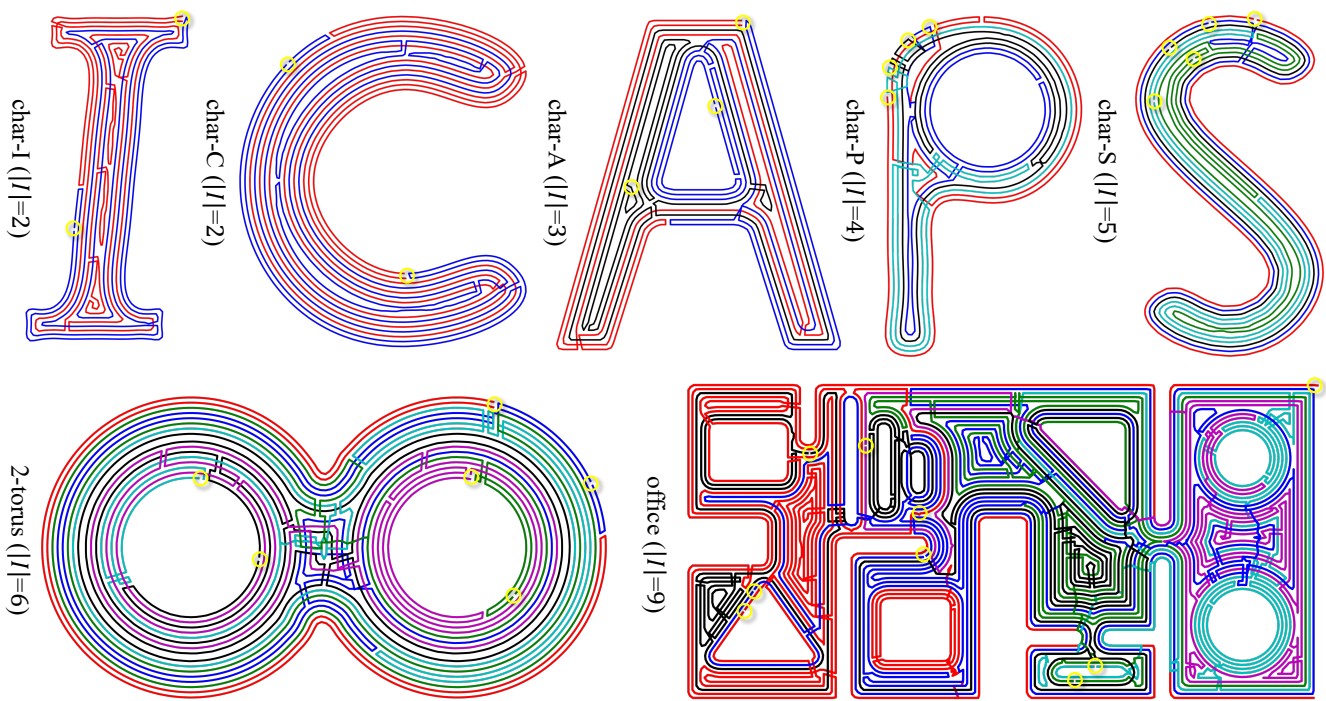

Figure 5: Coverage paths from MCFS. Different paths are in different colors. Yellow circles are root positions.

runtime limit of 30 minutes and an MST-based initial solution for warm start-up (Tang and Ma 2023). Whenever MCFS is equipped with isograph augmentation, the hyperparameter $\delta$ in Sec. 4.3 is set to $\min\{|I|, 4\}$, where $|I|$ is the number of robots for the MCPP instance, balancing between the MMRTC model complexity and the solution quality.

**Instances:** We conduct the experiments using MCPP instances displayed in Fig. 5, where the polygon workspace that needs to be covered is already filled with paths. The distance $l$ between adjacent isolines in all instances is $0.1$, which is also the cover diameter of the robots. The number of robots ($|I|$) of the instances range from 2 to 9. In instances *char-I* and *char-P*, two robots and four robots share the same root isovertex, respectively. In instance *2-torus*, three pairs of robots share three root isovertices, respectively. In all other instances, robots start from different root isovertices.

**Metrics:** In addition to the makespan $\tau$, we report the following metrics to evaluate an MCPP method and its solution: (1) Curvature: Average curvature of all paths (smaller values indicate smoother paths). (2) Coverage: Ratio between the covered area and the total workspace. (3) Overlapping: Ratio between the repeatedly covered area and the total workspace area. (4) Runtime: Total runtime of the method, including the MIP model solving time (when applicable).

**Stitching Tuple Selectors:** Tab. 1 compares curvature among the random, CFS, and MCS stitching tuple selectors. Both CFS and MCS selectors outperform the random selector, with average reductions of 24.6% and 27.9%, respectively. For less complex workspaces like *2-torus* that can be filled with smooth isolines, the CFS selector with staircase-like stitching paths outperforms the MCS selector since the

MCS selector struggles to distinguish small curvature differences. However, for complex workspaces like *office*, the MCS selector significantly excels by strategically selecting sharp corner points as stitching tuples, thereby substantially reducing the curvature. Based on these findings, the MCS selector will be used in the MCFS framework for the remainder of our experiments.

**Ablation Study:** To validate the effectiveness of isograph augmentation (Aug) and MMRTC solution refinement (Ref) presented in Sec. 4.2 and Sec. 4.3, respectively, Tab. 2 reports results for four MCFS variants: using only the original MMRTC solution, with Aug, with Ref, and combining both (labeled **NONE**, **AUG**, **REF**, and **BOTH**, respectively). Compared to NONE, REF and AUG reduce the makespan by an average of 29.7% and 36.0%, respectively. For instances *char-I*, *char-P*, *2-torus*, and *office*, this reduction is attributed to decreased overlapping ratio, particularly where the robot root positions are identical or adjacent. BOTH further enhances this effect in more complex instances for more complex instances like *2-torus* and *office*, doubling the reduction in the overlapping ratio, resulting in greater makespan reduction. For instances *char-C*, *char-A*, *char-S* where overlapping ratios of NONE are already low, the makespan reduction of REF results from the iterative cost-balancing procedure, whereas the makespan reduction of AUG results from a larger MMRTC solution space via the augmented edges. Although both REF and AUG require a longer runtime, this increase in runtime is less pronounced for complex instances where the MMRTC MIP model solving dominates. Overall, BOTH yields the largest average makespan reduction of 43.6% compared to NONE, combining the strengths

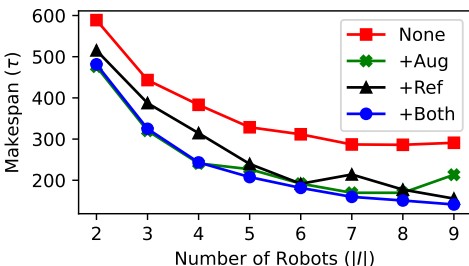

Figure 6: MCFS comparison on instance *office* with different number of robots.

of both REF and AUG in makespan minimization at the cost
575 of slightly longer runtime. Fig. 6 further shows the evolving
performance of four MCFS variants for *office* with increas-
ing numbers of robots and unique roots. It indicates that Ref
is more crucial with more robots because each robot needs to
cover fewer isolines, often leading to imbalanced MMRTC
580 trees, making isovertex splitting more effective in cost bal-
ancing. Aug consistently aids in reducing makespan by ex-
panding the MMRTC solution space, though it increases the
complexity and runtime of the resulting MIP model. The fig-
ure also shows that for $|I| \geq 7$, the MIP models become too
585 complex for all four variants to obtain satisfactory MMRTC
solutions within the runtime limit, whereas Ref, employed
by REF and BOTH, continues to significantly reduce the
makespan of the resulting suboptimal MMRTC solutions.

**Comparison:** We compare MCFS (+BOTH) with two state-
590 of-the-art grid-based MCPP methods, TMC (Vandermeulen,
Groß, and Kolling 2019) and TMSTC* (Lu et al. 2023),
that minimize path turns. To adapt TMC and TMSTC*
to the non-rectilinear workspaces in our instances, we use
overlay grids to approximate the workspaces, followed by
595 shortest pathfinding for robot return to root positions post-
coverage. Note that the reported coverage and overlapping
ratios for TMC and TMSTC* are approximations due to
the workspace approximation and small intersection of their
coverage paths with obstacles, whereas the values for MCFS
600 are exact. In Tab. 2, while the average coverage ratios of
TMC, TMSTC*, and MCFS are comparably close (with a
3.51% variance), MCFS demonstrates an average makespan
reduction of 32.0% and 27.9%, curvature reduction of 75.7%
and 47.8%, and overlapping ratio reduction of 13.6% and
605 20.9% compared to TMC and TMSTC*, respectively. Both
MCFS and TMC require longer runtime due to solving MIP
models for MMRTC and MTSP, respectively, especially in
instances with larger isographs or more robots, such as *2-
torus* and *office*). Fig. 7 and Fig. 8 visualize the coverage
610 paths planned by TMC and TMSTC*, respectively. These
paths exhibit a back-and-forth boustrophedon pattern, lead-
ing to high curvature and imperfect coverage around com-
plex obstacles. In contrast, MCFS notably excels in gener-
ating smooth paths that efficiently contour around arbitrar-
615 ily shaped obstacles, a clear visual advantage over the other
methods as shown in Fig. 5.

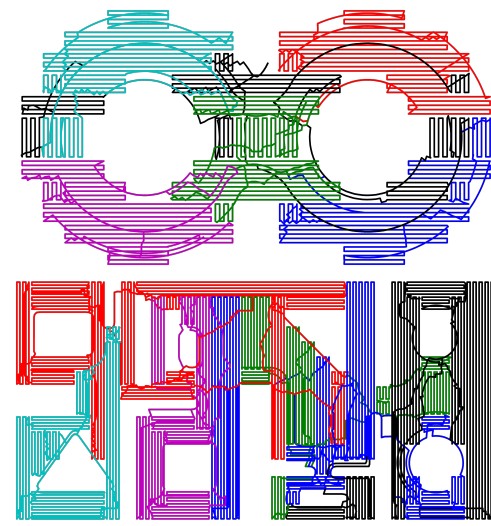

Figure 7: TMC MCPP solutions of *2-torus* and *office*.

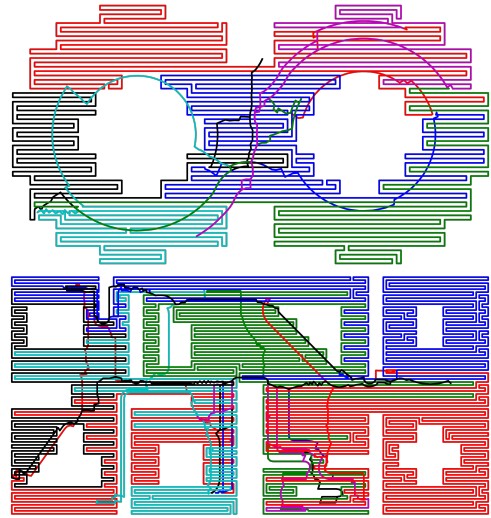

Figure 8: TMSTC* MCPP solutions of *2-torus* and *office*.

## 6  Conclusions

We proposed the MCFS framework, an innovative approach
that blends principles from computer graphics and auto-
mated planning to tackle the challenges of covering arbi- 620
trarily shaped workspaces in complex MCPP tasks. MCFS
leverages our novel unified version of CFS to bring scala-
bility and versatility for multi-robot scenarios by comput-
ing multiple rooted trees that jointly cover an input graph
of isolines. We also developed two effective optimization 625
techniques that significantly enhance the solution quality.
We validated the effectiveness of MCFS in various scenar-
ios through rigorous experimentation and analyses, bench-
marked against state-of-the-art MCPP methods. Future work
includes improving isoline generation to further boost the 630
coverage ratio, developing heuristics to accelerate the PIS
function for large numbers of robots or isolines, and speed-
ing up MMRTC solving.

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
