# OpenReview forum: "Multi-Robot Connected Fermat Spiral Coverage"
_icaps-conference.org/ICAPS/2024/Conference — ICAPS 2024_

### Official Review · Reviewer_XLb4 · 2023-12-26

**Significance And Importance:** 3
**Soundness:** 3
**Novelty:** 3
**Clarity:** 3
**Overall Evaluation:** 2
**Confidence:** 3

**Weaknesses:**

1: Minor weaknesses that are easily fixable.

**Contributions Of The Paper:**

The main contribution is to approach Multi-robot Coverage Path Planning (MCPP) using and adapting Coverage Fermat Spiral (CFS) to decompose the workspace into a set of paths for agents to follow. Necessary helping contributions are generalizing/unifying CFS, and designing refinement/improvement techniques for their MCFS solution.

**Ethical Considerations:**

(1) Not Applicable: The paper does not have any ethical considerations to address

**Nomination For Best Paper:**

No

**Questions For Authors:**

Major questions: None

Minor questions:
1. How would this method work for robots with kinematic constraints? I.e. if robots have a minimum turning radius, would MCFS still be applicable or would other machinery be required?
2. The isoline decomposition causes agents to cover long neighboring corridors of regions, as opposed to other approaches which generally have agents cover more compact regions. I.e. for the I in ICAPS (cute example by the way), the two agents have these long sections next to each other, when other approaches would generally split it to a top and bottom half. I am curious if your refinement process could work on arbitrary "isolines" that cover the region? I.e. instead of using CFS which produces longer neighboring isolines, other approaches could produce more compact isolines which could then be used in your method. Is this relevant future work that generalizes this idea or is CFS required?

**Reproducibility:**

4: Authors promise to release code and domains (whichever apply).

**Strengths Of The Paper:**

1. Upon a literature review using (Almadhoun et al. 2019), it seems that using computer graphics techniques is rarely used for multi-robot coverage. Instead, most approaches use standard graph decomposition techniques (e.g. grids/cells). This paper makes a convincing argument for looking into computer graphics techniques for graph decomposition for coverage.
2. Methods seem reasonable, the refinement processes make a sizable impact on performance.
3. Qualitative and quantitative results looks strong.

**Weaknesses Of The Paper:**

1. The paper is a tough read for non-computer graphics readers. I have traditional MAPF experience as well as some single-agent coverage experience but found the methods section hard to understand. I, and I think other readers, would really appreciate a beginner supplementary material that walks through fundamentals of CFS. One example is "Branch-and-cut-and-price for multi-agent path finding" which discusses a Branch-Cut-Price integer programming approach for MAPF. They specifically include supplementary material for MAPF readers unfamiliar to the general integer programming approach in the beginning of their methods section. I.e. look at Section 4 in  https://www.sciencedirect.com/science/article/pii/S0305054822000946#sec4 which refers the reader to https://ars.els-cdn.com/content/image/1-s2.0-S0305054822000946-mmc1.pdf which serves as a great primer.  I think including something similarly would really help the readability of the paper.
1b. Methods were hard to understand although the corresponding paragraph text helped to an extent. "PAIRWISEISOVERTICESSPLITTING" in Algorithm 2 is particularly tough to parse and I am unsure if anyone can understand it without the paragraph. Even with the paragraph understanding what is going on is hard.
1c. I did not understand Figure 3 at all. I found Figure 4 to be immensely helpful on the otherhand.

---

> ### Author Rebuttal · Authors · 2024-01-27
>
> **Weakness1**
> We highly agree with you on adding more CFS content to enhance readability. We will add an appendix in our technical report, although it would mostly reiterate concepts from the original CFS paper.
>
> **1b** We will refine the text to succinctly convey these concepts: Given an MMRTC solution, Alg. 2 aims to further reduce the makespan by iteratively segmenting an isoline and reassigning segments to trees/robots via PIS and AIR calls. Splitting vertex $v$ (1) reduces redundant coverage and (2) alleviates a high-cost tree covering $v$. AIR adds vertices to low-cost trees to encourage cost redistribution from high-cost trees. The vertex splitting in PIS operates on not only $v$ but also a neighboring $u$, essential for forming a closed loop-like path. PIS essentially selects the (heuristically) cost-minimal way of splitting $u$ and $v$, reassigns their segments to trees, and ensures validity.
>
> **1c** We will make Sec. 4.2 text more coherent with Fig. 3: It demonstrates adding the augmented edge $(v_1,v_3)$ with three valid stitching tuples in $O_{v_1,v_3}$ (represented by black boxes) and how $p_1$ and $p_3$ can be connected via $p_2$.
>
> **Q1** Like other MCPP methods, MCFS addresses back-end path planning (rather than front-end control) without explicitly considering kinematic constraints, such as minimum-turning radius, though its inherent attention to curvature partially addresses real-robot navigation feasibility. Future work could explore integrating such constraints into isoline generation and stitching within the MFCS framework.
>
> **Q2** Your question indeed opens up an interesting direction. While untested, alternative methods could generate more compact isolines (or other types of space-filling lines) or ones respecting kinematic constraints as suggested above. In general, MCFS is compatible with any resulting isograph that aligns with our definition, regardless of its topology.
>
> **Below used for rebuttal to aseZ (2/2)**
>   - Optimizing continuity and smoothness, alongside makespan minimization, enhances real robot navigation, aligning with standard practices in robotics and control. The advantage of MCFS w.r.t. these metrics is experimentally validated, although we acknowledge the theoretical challenge of achieving optimality for these metrics simultaneously.
>   - As rigorously defined in Eq. (6) and elaborated thereafter, $E$# is the set of augmented edges to be added to the isograph to reduce its sparsity and improve MMRTC solutions.

---

### Official Review · Reviewer_aseZ · 2023-12-28

**Significance And Importance:** 1
**Soundness:** 2
**Novelty:** 2
**Clarity:** 2
**Overall Evaluation:** 1
**Confidence:** 4

**Weaknesses:**

-1: Major weaknesses requiring significant work to be addressed for the paper to be accepted.

**Contributions Of The Paper:**

This paper solves the problem of for Multi-Robot Coverage Path Planning (MCPP) using a previously proposed concept called Connected Fermat Spiral (CFS). Some improvements have been made, including extending the previously unidirectional graph to bidirectional graph, and extending the single robot planning to multi-robot scenarios.

**Ethical Considerations:**

(1) Not Applicable: The paper does not have any ethical considerations to address

**Nomination For Best Paper:**

No

**Questions For Authors:**

•	In abstract:
-	You define CFS as Coverage Fermat Spiral here. But the title of Section 3 is Connected Fermat Spiral (CFS). Please double check and make it consistent.
•	In introduction:
-	Please clarify what do you mean by ‘more organic and less segmented coverage’.
-	‘adeptly’ is a typo.
-	‘Key Contributions: …’, it’s a weird way of starting a new paragraph.
•	In Related Work:
-	‘interesting readers’ is a typo.
-	‘geometrical critical’ is a typo.
•	In Connected Fermat Spiral (CFS):
-	It is better to first define graph structure.
-	It is unclear how vertices can represent an isoline? Do you mean dividing an isoline into a sequence of waypoints? Then, how do you express the sequence/order/association of the waypoints on an isoline? A conventional graph <V, E> does not express such information. Please clarify.
-	It is unclear what are adjacent segments.
-	It is unclear what is a pocket. What are the components you are referring? Please clarify.
-	Please avoid citing Wiki. Use a proper citation instead.
-	‘Rather than explicitly identifying …. Instead of explicitly identifying …. ‘ it seems two redundant sentences here. Please double check.
-	What do you mean by ‘layered isolines’? I guess you mean isolines on different levels. Please double check and clarity.
-	Please clarify how do you sampling a 2D mesh grid of points within the polygon. For instance, is it uniform sampling in each dimension?
-	What is a distance step size? Each layer should be each level.
-	It is hard to understand Eq. (1). if Lz = Lv, Iz and Iv would overlap (one isoline). Why d(p, Iv) < d(p, Iz)? This equation does not make sense to me, please clarify.
-	It is unclear to me what it means if Ouv is non-empty. Please clarify.
-	‘It also avoids adding edges (u, v) where the respective isolines Iu and Iv are separated by multiple isolines’ it is also unclear to me how this can be achieved by Eq. (1), please clarify.
-	The logical and symbol in Eq. (2) is improper here. Just use a comma.
-	Eq. (2) also does not make sense to me. If p is uniquely defined as the nearest point on Iu to q, (p, q) should just be a fixed point. What is the point defining a space Qu-v \times Qv-u?
-	‘Subsequently, an undirected edge (u, v) is formed for any u, v ∈ V in adjacent layers with a nonempty Ou,v.’ Since Eq. (1) and (2) are not clear to me, it is hard to see what this means and how it is achieved.
-	‘connect isolines Iu and Iv by stitching p to q and Bu(p) to Bv(q), where Bu(p) denotes the point preceding p along isoline Iu in counterclockwise order.’ It is unclear to me
1)	how connecting Iu and Iv is achieved by stitching p, q, Bu(p), Bv(q).
2)	why stitching Bu(p) to Bv(q)? Is it for continuity? Then why stitching only one preceding point? Is it better to stitch more points for continuity?
3)	if for other purposes, please clarify.
-	CFS (Zhao et al. 2016) you just need to cite this paper once the first time you mention it.
-	It seems to me ‘versatile’ means the starting point can be on a random level, instead of the lowest level. If I’m right, please state clearly. Otherwise, please clarify what versatile means.
-	In Fig.1, all figures start from the lowest level. Can you give some examples where the starting point is on a random level?
•	In Multi-Robot CFS Coverage:
-	It is unclear to me how the authors address task allocation, cost distribution, and cooperation. Task allocation and cost distribution are mentioned as advantages of their method in Introduction. Cooperation is necessary for multi-robot operations.
-	If Eq. (4) is the objective for multi-robot operation, the robots only find the paths that minimize the overall cost. Then, how do they achieve task allocation and cost distribution?
-	‘Π, the path length is linear in |π| and thus the cost of any path π can be evaluated as c(π) = |π|’ If the cost is just path length, what is the point optimizing continuity and smoothness?
-	It is unclear to me what E# represents.
•	In Empirical Evaluation:
-	I can’t see any evaluation on energy consumption (total or balanced). Perhaps the path curvature can somehow reflect the energy consumption, it needs to be rigorously defined.

**Reproducibility:**

4: Authors promise to release code and domains (whichever apply).

**Strengths Of The Paper:**

The paper is generally readable.

**Weaknesses Of The Paper:**

•	The idea and algorithms are not clearly presented, and are hard to understand.
•	The equations are not well defined and some parts do not make sense.
•	The experiments are not well designed. Their claims on energy consumption are not supported by their experiments.

---

> ### Author Rebuttal · Authors · 2024-01-27
>
> **General/Weakness1&2** Writing style choice: Our sections start with an overview before detailing technical aspects. Concepts initially left undefined/unexplained are rigorously detailed in later technical subsections.
>
> **Evaluation/Weakness3** Our evaluation's makespan metric captures energy consumption. Reduced makespan typically implies decreased operational time and energy for multi-robot systems. We will use "operation cost" if "energy" is considered unsuitable.
>
> **Abstract/Introduction/Related Work**
>   - We will fix all typos. CFS should consistently denote Connected Fermat Spiral.
>   - "More organic and less segmented coverage" implies that MCFS uses layered isolines for coverage without explicit decomposition, offering a 'less segmented' approach. It allows MCFS paths to seamlessly adapt to any workspace shape, as shown when contrasting Fig. 5 and Fig. 7, providing a more 'organic' fill (a term we intend to refine).
>
> **CFS**
>   - Writing style choice: We outline concepts in the first paragraph before formally defining the isograph $G=(V,E)$, where each vertex is associated with a unique isoline (also see Fig. 4). *Adjacent segments* are formally defined in Eq. (2) as set $O_{u,v}$ using *waypoints* on isolines and determine vertex adjacency. The above concepts are rigorously defined in Sec. 3.1, though we intend to make them clear further.
>   - *Pockets* are a spanning tree's connected components (already defined in Sec. 3). Note: Unlike (Zhao et al. 2016), our unified CFS obviates explicit pocket construction, simplifying the process.
>   - You are correct: *Layered isolines* are isolines in different layers, and *sampling ... polygon* is uniform sampling per dimension. *Distance step size* is the constant distance between isolines in adjacent layers.
>   - Note that multiple isolines may coexist within one layer as we'll exemplify with Fig. 4 (yellow and green isolines). $I_z$ and $I_v$ are different despite sharing a layer ($L_z=L_v$) since each vertex is associated with a unique isoline per our definition.
>   - Eq. (1) is very clear: $O_{u\rightarrow v}$ contains points on $I_u$ closest to $I_v$ among isolines one layer away, but not those points closest to any other $I_z$ in the same layer as $I_v$. Typo: $O_{u,v}$ on Line 220 should be $O_{u\rightarrow v}$.
>   - A nonempty $O_{u\rightarrow v}$ means that $I_u$ and $I_v$ are connectable via points in the set, forming edge $(u,v)$.
>
> **Continue at the end of the rebuttals to b9n5 and XLb4 due to limit**

---

### Official Review · Reviewer_b9n5 · 2024-01-24

**Significance And Importance:** 2
**Soundness:** 3
**Novelty:** 2
**Clarity:** 3
**Overall Evaluation:** 1
**Confidence:** 2

**Weaknesses:**

2: No major or minor weaknesses.

**Contributions Of The Paper:**

The paper presents a multi-robot connected fermat spiral algorithm for addressing multi-robot coverage path planning problems.
The work leverages results from computer graphics adapting Coverage Fermat Spiral /CFS) solution.

**Ethical Considerations:**

(5) Excellent: The paper comprehensively addresses all of the applicable ethical considerations

**Nomination For Best Paper:**

No

**Questions For Authors:**

Is there any MCPP benchmark to be considered for evaluating the effectiveness of the approach?

Is the cost for encoding MCPP negligible? Or should this be a further cost to be considered in the evaluation?

**Reproducibility:**

3: Authors describe the implementation and domains in sufficient detail.

**Strengths Of The Paper:**

The presents a nice and original solution for the MCPP problems leveraging a solution from a completely different field.io

**Weaknesses Of The Paper:**

The experimental analysis seems to consider specific scenarios but there is no reference to benchmarks.

---

> ### Author Rebuttal · Authors · 2024-01-27
>
> **Weakness/Benchmark**
> Existing MCPP benchmarks, like those in (Tang et al. 2023), are tailored for grid-based methods on 2D grid maps. We assert that the instances used in our paper (Fig. 5) are diverse, ranging from fully non-rectilinear (2-torus) to mostly rectilinear (office) ones. Our evaluation of effectiveness is thorough, including diverse metrics not typically used in MCPP research.
>
> **Encoding cost**
> The computation cost for encoding MCPP is indeed small yet not negligible. It includes generating the layered isolines and constructing the MMRTC model. Both are already counted in the reported runtime metric in Tab. 2.
>
> **Below used for the rebuttal to aseZ (1/2)**
>   - A nonempty $O_{u\rightarrow v}$ means that $I_u$ and $I_v$ are connectable via points in the set, forming edge $(u,v)$.
>   - Obviously, the construction does not connect non-adjacent-layer isolines. Moreover, Fig. 4 clearly shows how Eq. (1) avoids connecting adjacent-layer isolines (e.g. gray & yellow) without a nonempty O set, separated by other isolines.
>   - We agree that a comma is proper but assert that Eq. (2) is mathematically precise which indicates $(p,q)\in O_{u,v}\subseteq Q_{u\rightarrow v}\times Q_{v\rightarrow u}$. Specifically, $O_{u,v}$ simultaneously draws $p$ from $Q_{u\rightarrow v}$ and $q$ from $Q_{v\rightarrow u}$ that satisfy being closest to each other, rather than drawing $p$ first and then setting $q$ as the nearest point on $I_v$ to $p$.
>   - "Subsequently,...": clarified above.
>   - "connect ...": The 4 points $p,q,B_u(p),B_v(q)$ are used to create a closed path from two closed isolines $I_u,I_v$ (squares in Fig.4 represent these points). The design choice of stitching only one preceding (or succeeding) point makes sure each point is covered, a concept derived from the original CFS work.
>   - We acknowledge that "versatile" means being able to start from any layer and will update Fig.1 accordingly. Yet, Fig. 5 already shows such examples where starting points are not exclusively on boundaries.
>
> **MCFS**
>   - MCFS addresses the three concepts via offline centralized planning. Task allocation: Computing trees rooted in robot starting points. Cost distribution and cooperation: [1]MMRTC's makespan minimization (Eq. (4)) on a weighted isograph distributes nonuniform-cost isolines to robots. [2]Our two optimizations expand the solution space and allow distributing single isoline coverage among multiple robots.
>
> **Continue at the end of the rebuttal XLb4 due to limit**

---

### Meta-Review · Area_Chair_agn4 · 2024-02-06

**Recommendation:** Accept (Oral)
**Confidence:** 3

**Metareview:**

The paper studies a variant of the multi-robot coverage problem and suggests using specific techniques from computer graphics to solve it. This is an interesting direction and all reviewers acknowledge that the work is likely to provide value to the ICAPS community as the presented method is original and addresses several challenges that can not be easily handled within the commonly used approaches (like the ones based on the cellular decomposition).

The main concern is the clarity. The ideas behind the algorithms are not clearly presented and are hard to understand. It is highly recommended to enhance the presentation (e.g. by providing more illustrative examples and a beginner material that walks through the fundamentals of CFS).

**Ethical Considerations:**

(1) Not Applicable: The paper does not have any ethical considerations to address